# Enhancing Question Answering on Charts Through Effective Pre-training Tasks

**Ashim Gupta**[1,†,*] **, Vivek Gupta**[2]**, Shuo Zhang**[3]**,**
**Yujie He**[3]**, Ning Zhang**[3]**, Shalin Shah**[3,‡]
[1]University of Utah, Salt lake, UT
[2]University of Pennsylvania, Philadelphia, PA
[3]Bloomberg, New York, NY
[†]ashim@cs.utah.edu, [‡]sshah804@bloomberg.net

## Abstract

To completely understand a document, the use of textual information is not enough. Understanding visual cues, such as layouts and charts, is also required. While the current state-of-the-art approaches for document understanding (both OCR-based and OCR-free) work well, we have not found any other works conducting a thorough analysis of their capabilities and limitations. Therefore, in this work, we address the limitation of current VisualQA models when applied to charts and plots. To investigate shortcomings of the state-of-the-art models, we conduct a comprehensive behavioral analysis, using ChartQA as a case study. Our findings indicate that existing models underperform in answering questions related to the chart's structural and visual context, and also numerical information. To address these issues, we propose three simple pre-training tasks that enforce the existing model in terms of structural-visual knowledge, and its understanding of numerical questions. We evaluate our pre-trained model (called MatCha-v2) on three chart datasets - both extractive and abstractive question datasets - and observe that it achieves an average improvement of 1.7% over the baseline model.

## 1 Introduction

Understanding and extracting insights from charts and plots is a fundamental aspect of data analysis that is critical for various domains, including finance, healthcare, and scientific research. To bridge the gap between raw data and actionable knowledge, question answering (QA) systems tailored for charts and plots have gained increasing attention in recent years. These systems aim to enable users to pose natural language questions about the content of visual data representations, such as bar graphs, line charts, and scatterplots, and receive informative answers. However, despite the remarkable progress made in developing such QA systems, there remains a significant challenge: their performance often falls short when subjected to human-generated questions. This discrepancy between machine performance and human expectations underscores the need for a comprehensive investigation into the limitations of existing models and the development of strategies to address their shortcomings.

To shed light on the shortcomings of current chart-based QA systems (Masry et al., 2022), this paper first undertakes a detailed checklist-based behavioral analysis (Ribeiro et al., 2020; Bhatt et al., 2021; Rogers et al., 2021). Choosing the models trained on the ChartQA dataset (Masry et al., 2022) for the representative case study, we systematically evaluate model responses against examples constructed from the checklist of expected behaviors, allowing us to pinpoint the areas in which these models falter. The checklist is designed to assess various dimensions of chart-based question answering, including the ability to interpret the structural and visual context of the chart, handle questions requiring numerical reasoning, and offer meaningful insights that align with human expectations. Concretely, our analysis reveals that current state-of-the-art models perform poorly on two types of questions. The first type of questions are those that pertain to the visual aspects of a chart (e.g., color), while the second type are questions that require application of numerical operators to numerical items present in the chart (e.g., average, etc.).

To address these two shortcomings, we propose a set of three pre-training tasks: Visual-Structure prediction, Summary Statistics prediction, and Numerical Operator prediction. Through evaluation on three chart question answering datasets, we find that models fine-tuned after using this pre-training outperform the baseline model by more than 1.7 percentage points in an absolute sense.

In summary, our contributions are: 1) We per-

---

[*] Work done during summer internship at Bloomberg.

form a checklist-based behavioral analysis of the current state-of-the-art chart question answering systems to identify issues and challenges faced by such systems. 2) We propose three simple, yet effective, pre-training tasks to address these issues. The new pre-trained model outperforms the baseline systems significantly.

## 2 Behavioral Analysis via Checklist

We first describe the procedure for constructing the checklist for chart-based question answering systems and then discuss the results and observations.

### 2.1 Checklist For ChartQA

To perform the perturbation analysis for the chart-based models, we choose the popular ChartQA dataset, where the objective is to answer questions based on the information provided in the accompanying charts. This real-world dataset consists of four (4) different subsets: (a) Statista-H, (b) Pew, (c) Our World In Data (OWID), and (d) OECD. However, in all our checklist analysis and evaluation, we only use one subset of data, namely OWID, as the library (owid-grapher) to generate the charts and apply perturbations is available.[1] All the other data subsets were either manually curated or hard-coded and hence cannot be perturbed. To construct the checklist for behavioral analysis, we leverage 400 different charts and the corresponding data points sourced from Our World In Data. To ensure the accuracy and reliability of our analysis, we manually design three distinct types of templates: Structural & Visual, Data Extraction, and Numerical QA.

**Structural and Visual** The Structural & Visual templates are crafted to assess the model's understanding of chart structures and visual elements. For instance, we evaluate whether the model can discern between different types of charts, such as bar charts or line charts, and if it can recognize various colors used in the charts.

**Data Extraction** The Data Extraction templates gauge the model's ability to accurately retrieve data values from the charts.

**Numerical QA** Last, the Numerical QA templates are tailored to assess the model's proficiency in answering both simple and complex numerical questions related to the data points present in the charts.

---

[1] https://github.com/owid/owid-grapher-py

In total, we have 33 distinct QA templates belonging to these three classes. A few examples are shown in Table 1 and detailed examples are presented in the Appendix in Table **??**.

### 2.2 Evaluation and Results

As mentioned earlier, our focus is to perform behavioral analysis using the checklist for the chart-based models on the ChartQA dataset to first understand their extent of chart understanding and then pinpoint areas where these models require improvement.

**Models Evaluated** Our focus in this work entails the two large, recently introduced chart pre-trained models in MatCha, and DePlot + LLM. While MatCha is an end-to-end chart-to-text pre-trained model, DePlot + LLM is a pipelined approach where DePlot first converts an input chart into its textual representation in the form of a table and then performs few-shot question answering via a Large Language Model (LLM). In our experiments, we use the FLAN-UL2 (Tay et al., 2022) with 20 billion parameters as the LLM for DePlot + LLM evaluation. We do not evaluate models such as VisionTapas (Masry et al., 2022), Vl-T5 (Cho et al., 2021) since they are harder to work with than the two models we use.

**Evaluation Metric** For each of the templates we use, we measure the model's *Failure Rate*, i.e., the number of examples where model prediction does not match the expected/gold output. We present examples of some failure cases along with failure rates in Table 1.

**Results** The results are shown in Table 1. Notably, the Structural & Visual templates exhibit alarmingly high failure rates, prompting an in-depth investigation into the underlying causes. One plausible explanation for these high failure rates is the absence of explicit enforcement of structural and visual information in the pre-training tasks for models like MatCha and DePlot. This absence underscores a critical gap in the models' understanding of fundamental chart structures and visual elements, posing a significant challenge in their interpretation of complex data visualizations.

In stark contrast, the models demonstrate a significantly higher proficiency in handling templates focused on data extraction. The lower failure rates observed in this category highlight the models' capability to accurately extract data points from

| Chart Capability & Template Description | | Failure Rate (%) | | | |
|---|---|---|---|---|---|
| | | MatCha | DePlot | MatCha-v2 | DePlot-v2 |
| **Structural & Visual** | **Colors in Chart**: Is a certain color present or absent? | 98.9 | 99.5 | 15.6 | 23.5 |
| | **Chart Type**: Is it bar plot or a line plot? | 99.4 | 74.9 | 2.2 | 8.4 |
| **Data Extraction** | Extract Entity Name from Original Chart (a) | 33.9 | 1.6 | 31.2 | 1.8 |
| | Extract Entity Value from Original Chart (a) | 63.5 | 0.8 | 25.6 | 1.1 |
| | Extract Value from Perturbed Chart - Sort Descending Order (b) | 13.3 | 1.6 | 12.4 | 1.5 |
| | Extract Value from Perturbed Chart - Add Irrelevant Bar (c) | 35.0 | 1.1 | 31.2 | 1.1 |
| **Numeric Q/A** | **Operator**: Sum | 99.5 | 45.8 | 62.3 | 44.5 |
| | **Operator**: ArgMax | 16.3 | 15.2 | 11.5 | 16.1 |
| | **Operator**: Average + Comparison | 83.4 | 65.2 | 47.5 | 61.0 |

Table 1: A partial selection of Template based tests for the ChartQA using our checklist. We report Failure Rate (in %) for each of the templates. The proposed v2 versions significantly decrease the failure rate. Please refer to Table **??** in the Appendix for the examples of each type of template.

diverse charts, indicating a relatively robust performance in tasks requiring precise information retrieval.

We also notice intriguing disparities in the models' performance concerning numerical QA templates. While the failure rates soar for complex mathematical operations, such as intricate calculations involving multiple operators, a marked improvement is observed in tasks involving simpler operations like finding maximum or minimum values. This nuanced discrepancy suggests that while the models struggle with advanced numerical reasoning, they exhibit a more stable grasp on elementary mathematical concepts. This observation not only sheds light on the specific challenges faced by these models in handling complex mathematical operations but also underscores their potential strengths in addressing simpler, more straightforward numerical queries.

## 3 Experiments

### 3.1 Proposed Pre-training Tasks

As discussed earlier, we proposed a comprehensive approach to address the challenges identified through checklist-based analysis. In this section, we introduce three distinct pre-training tasks designed to enhance the MatCha model's chart understanding capabilities.

**Visual Structure Prediction** The first task, termed Visual Structure Prediction, demands the model predict intricate details of input charts, encompassing chart types (such as bar, line, etc.),

colors associated with chart entries, and even chart titles.

**Summary Statistics Prediction** The second task, Statistics Prediction, focuses on refining numerical question-answering by training the model to predict summary statistics like mean, maximum, and minimum values from the chart data.

**Numerical Comparison** Finally, the numerical comparison task requires the model to compare values from different chart entries and predict relationships such as *greater than*, *smaller than* or *equal to*. By engaging in these tasks, the MatCha model undergoes targeted pre-training to bolster its chart comprehension abilities, paving the way for more accurate and insightful responses in question-answering scenarios.

An example for two of these three pre-training tasks is shown in fig 4.

**Extension to DePlot** Since DePlot is a chart-to-table generation model, we only pre-train the De-Plot model on the first two pre-training tasks of Visual Structure Prediction and Summary Statistics Prediction.

**Pre-training Details** We pre-train both models on the charts extracted from the training data of the ChartQA dataset. We continue pre-training from the initial pretrained variants and pre-train each model for only one epoch as we saw significant reduction in accuracy on the validation set. In addition, we use batch size of six (6) and use all the other hyperparameters as suggested by (Liu

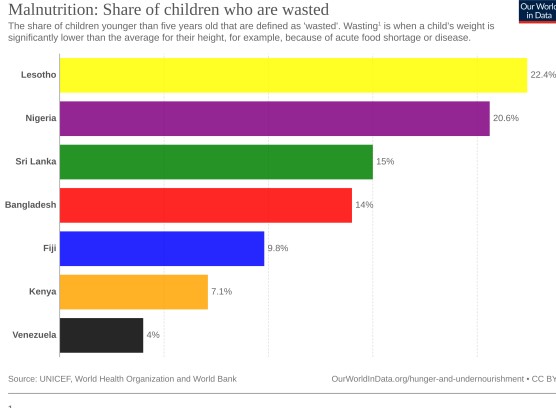

Figure 1: An unperturbed chart without any modifications from the ChartQA dataset. The chart comes from the OWID website. Our aim is to perturb these charts and evaluate models using our proposed checklist.

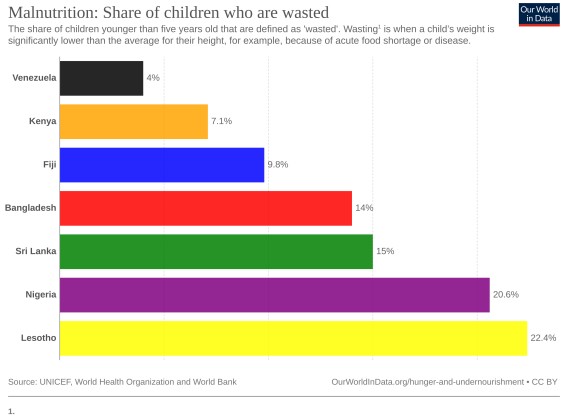

Figure 2: A perturbed chart where the bars are sorted in descending order. We find that the models (especially MatCha) are sensitive to the order in which the bars appear.

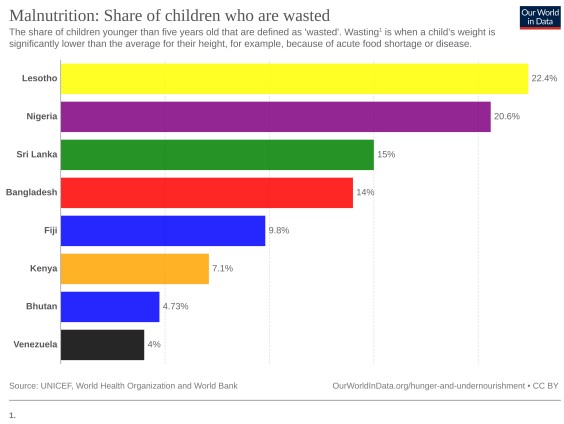

Figure 3: A perturbed chart where a bar unrelated to the question is added. Adding an unrelated entity/bar to the chart reduces performance.

et al., 2023a). We denote our proposed versions of MatCha and DePlot with the v2 suffix.

**Implementation Details and Computing Infrastructure Used**   We use batch size of six (6) to train the MatCha models due to computational constraints. Additionally, all the models that require training (e.g., MatCha) were trained up to five epochs. All of our experiments required access to GPU accelerators. We ran our experiments on two machines: NVIDIA Tesla V100 (16 GB VRAM) and Tesla V100 (32 GB VRAM). We did not experiment with VisionTapas (Masry et al. (2022)) as we could not run the publicly released implementation due to a missing dependency. [2] We train all our models using the Transformers library from Hugging Face (Wolf et al., 2019) with the PyTorch back-end (Paszke et al., 2019).

**Improved QA on Checklist**   We first measure the effectiveness of these two pre-training tasks on the proposed checklist. As shown in Table 1, the proposed variations of the two models (called v2 versions) significantly decrease failure rates across all template and question types.

## 3.2   Chart Question Answering Evaluation

To assess the effectiveness of our proposed pre-training tasks on actual question answering tasks, we conduct experiments on the ChartQA dataset. Furthermore, we extend our evaluation to a different question-answering dataset named PlotQA, wherein charts and associated questions are derived from disparate sources. This cross-dataset evaluation enables us to gauge the model's adaptability and generalizability beyond its original training data.

Both ChartQA and PlotQA are extractive question-answering datasets, where an answer is retrieved by combining entries from the chart. In addition, we explore the model's performance in abstractive question-answering, a more complex task necessitating detailed descriptive responses, using the OpenCQA dataset. Notably, both ChartQA and PlotQA focus on extractive question-answering, demanding precise extraction of information from the source data, while OpenCQA necessitates the generation of more extensive, contextually rich answers. Through these evaluations, we gain a holistic understanding of the MatCha model's capabilities, from basic chart comprehension to nuanced

---

[2]An issue on the github repository of the code base: https://github.com/vis-nlp/ChartQA/issues/9

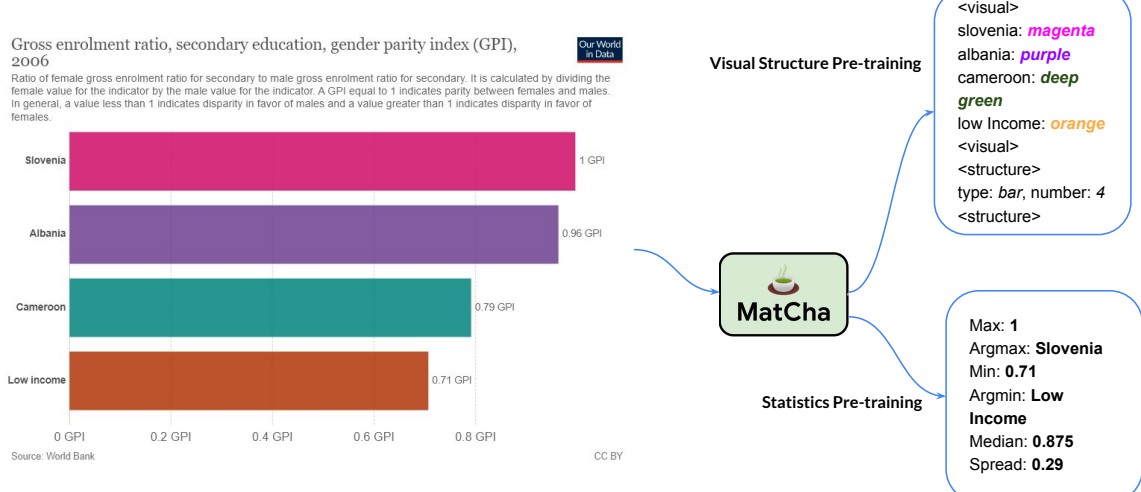

Figure 4: Two of the proposed pre-training procedures: Visual Structure prediction and Summary Statistics prediction. In the example shown, for the pre-training task involving visual structure prediction, the model is asked to predict the color of the bar corresponding to each entity as well as the structure and type of the chart. As shown, for the Summary Statistics prediction, the model has to output the statistics of the data shown in the chart (ex: Maximum, Median, etc.).

and elaborate question answering across diverse datasets and question types.

### 3.3 Evaluation Metrics and Datasets

**Datasets** We evaluate the effectiveness of our proposed pre-training methods on the three chart-related datasets. Here is a brief summary of each dataset, along with comments on how we use it to evaluate a model's chart understanding capabilities:

*Chart QA* The Chart Question Answering (QA) (Masry et al., 2022) dataset, as the name suggests, is a natural language query and answer generation dataset with one key difference from most NLP datasets – it also provides the visual representation of data or the chart image, which contain richer information such as layout, colors, etc. It contains approximately 28,000 training examples and comes with two evaluations sets: Augmented and Human. The Augmented set was constructed using a question generation system, while the Human set was made entirely from human annotations.

*Plot QA* The Plot Question Answering (QA) dataset (Methani et al., 2020) is also a visual question answering dataset based on real-world scientific charts and question-answer pairs collection from crowd-sourced templates. This dataset has 80% QA pairs whose answer is either not present in the chart or not in vocabulary, which means it con-

tains a nice breath of data variability. PlotQA contains approximately 5 million training examples. To reduce the computational burden, we sample approximately 25,000 of those randomly sampled examples for our evaluation.

*OpenCQA* The Open Chart Question Answering (CQA) dataset (Kantharaj et al., 2022) is designed specifically for open-ended question answering on charts. These questions span from summarizing the trend observed in the chart to describing and comparing a certain attribute over the period considered in the chart. OpenCQA contains approximately 7,200 training examples.

**Evaluation Metrics** For ChartQA and PlotQA, we follow the original authors and use relaxed accuracy (correct answer within a tolerance of 5%), while we use ROUGE metrics for OpenCQA, as it is a generative text-heavy dataset.

### 3.4 Results with Pre-training

As can be observed from Table 2, we see consistent improvements for the MatCha model across all three evaluation datasets. Perhaps surprisingly, we observe a much larger improvement for PlotQA than for the ChartQA dataset from which pre-training data was used. We also gain improvements for the OpenCQA dataset, which requires long-form question answering. The improvement for DePlot-v2 is smaller than that for MatCha-v2, as

| Model | ChartQA | | PlotQA | OpenCQA |
|---|---|---|---|---|
| | Aug | Human | | |
| MatCha | 77.0 | 28.7 | 52.0 | 29.19 |
| DePlot + Flan UL2 | 69.4 | 22.4 | 50.2 | **36.52** |
| MatCha-v2 (Ours) | **78.3** | **30.1** | **55.8** | **29.6** |
| DePlot-v2 + Flan UL2 (Ours) | **71.5** | **24.2** | **51.1** | 35.47 |

Table 2: Evaluation Results for the proposed pre-training methods. We measure accuracies for the ChartQA and PlotQA datasets, while ROUGE is used for the OpenCQA dataset. As can be observed, our proposed pre-training methods significantly improves MatCha model (called MatCha-v2) across all three datasets. We highlight the entries where proposed variation provides the improvement over the baseline.

the component responsible for question answering is the LLM that remains unchanged.

Since our pre-training procedure uses charts sourced from the ChartQA dataset, the evaluation on two other datasets forms an out-of-domain evaluation. Particularly on PlotQA, where charts are from different sources than those used in ChartQA, we see more significant improvements than ChartQA. Finally, although performance improvements are less significant on the OpenCQA dataset, this is not entirely unexpected, as the checklist we devised was for extractive QA. As such, the pre-training tasks motivated from those results are also more suited for extractive QA.

## 4 Related Work

Numerous studies have highlighted various robustness challenges in NLP models, including their over-sensitivity to minor perturbations (Ebrahimi et al., 2018; Wallace et al., 2019) as well as under-sensitivity to large changes (Gupta et al., 2021; Feng et al., 2018). A prominent method for identifying these issues is through behavioral analysis using specifically designed checklist examples (Ribeiro et al., 2020). In this work, we build upon this approach by applying it to multimodal chart-reasoning models, developing a checklist that aids in identifying similar robustness concerns.

Tackling these robustness challenges presents a distinct set of difficulties. Although increasing model size has resolved some of these issues (Gupta et al., 2024), this approach is not always ideal. In this study, we demonstrate that certain robustness problems can be mitigated by designing targeted pre-training tasks. For instance, we introduce a pre-training task where the model

predicts the visual structure of the input chart, which improves its ability to answer questions related to colors and plot types. Similar to our work, UniChart (Masry et al., 2023) explores pre-training a multimodal model to perform both low-level tasks (e.g., extracting visual elements) as well as high-level tasks (e.g., chart summarization).

## 5 Conclusion

In this work, we present a detailed study to evaluate the chart understanding capabilities of two current state-of-the-art QA models. We propose a detailed checklist that can be used to high- light the current shortcomings of these models. Broadly, we evaluate end-to-end QA models like MatCha and pipeline based models like DePlot + LLM. These help us identify avenues of improvement. Using them, we show that adding relevant pre-training tasks improves the performance of the model to achieve performance improvements across three datasets. Across the three tasks and datasets we consider, our pre-training methods help MatCha achieve an average improvement of 1.7% points.

## 6 Limitations

We foresee one main limitation of this work. We do not conduct checklist analyses and evaluation with the latest proprietary models like GPT-4o (Achiam et al., 2023), or Claude 3.5 Sonnet. The main reason for this is the cost and budget restrictions of our project due to the large number of checklist-based evaluation examples. Additionally, we do not experiment with the latest open-source Large Multimodal Models (LMMs) such as LLaVA-1.5 (Liu et al., 2023b) or LLaVA-NeXT (Liu et al., 2024) as we found them to underperform the task-specific chart models like MatCha.[3]

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

| | Chart Capability & Template Description | Typical Examples
**Q:** denotes the question, **G:** Gold, and **P:** Predicted |
|---|---|---|
| **Structural & Visual** | **Colors in Chart**: Is a certain color present or absent? | **Q:** Is there a bar of orange color in the given chart?
**P:** yes, **G:** yes |
| | **Chart Type**: Is it bar plot or a line plot? | **Q:** Is there a bar of blue color in the given chart?
**P:** line plot, **G:** bar plot |
| **Data Extraction** | Extract Entity Name from Original Chart (a) | **Q:** Which country has wasting percentage as 9.8?
**P:** Fiji, **G:** Fiji |
| | Extract Entity Value from Original Chart (a) | **Q:** What is the wasting percentage of Fiji in the given chart?
**P:** 3.8, **G:** 9.8 |
| | Extract Value from Perturbed Chart - Sort Descending Order (b) | **Q:** What is the wasting percentage of Fiji in the given chart?
**P:** 0.8, **G:** 9.8 |
| | Extract Value from Perturbed Chart - Add Irrelevant Bar (c) | **Q:** What is the wasting percentage of Fiji in the given chart?
**P:** 0.8, **G:** 9.8 |
| **Numerical Q/A** | **Operator**: Sum | **Q:** What is the sum of wasting percentage for Nigeria and Sri Lanka bars? **P:** 23.5, **G:** 35.6 |
| | **Operator**: ArgMax | **Q:** Which country has the highest wasting percentage?
**P:** Malnutrition, **G:** Lesotho |
| | **Operator**: Average + Comparison | **Q:** How many countries have wasting percentage more than average of all the countries? **P:** 2, **G:** 3 |

Table 3: A partial selection of Template based tests for the ChartQA using our checklist and corresponding examples. We also show the predictions from the MatCha model on each of these examples.

# A  Appendix

## A.1  Checklist Examples

We show the examples of the checklist tests presented in the main section of the paper in following table 3.