# OpenReview forum: "Enhancing Question Answering on Charts Through Effective Pre-training Tasks"
_EMNLP/2024/Workshop/BlackBoxNLP — BlackboxNLP 2024_

### Official Review · Reviewer_m1xT · 2024-09-05

**Overall Assessment:** 2
**Confidence:** 3

**Best Paper:**

1

**Best Paper Justification:**

N/A

**Comments Questions Suggestions And Typos:**

My main question is about finetuning versus pretraining-- see discussion above.

**Paper Summary:**

This paper focuses on question-answering from charts, and includes both an evaluation component and a pretraining experiment. It explores the limitations of two models, MatCha and DePlot + LLM, finding two common error areas: failures to understand visual properties of the charts and failures related to numerical reasoning. Based on these findings, the authors train MatCha and DePlot with three additional tasks to try to augment their abilities in these areas. These versions of the models demonstrate modest performance gains over the originals.

**Summary Of Strengths:**

I think the most interesting aspect of this paper is the checklist investigation of the limitations of models on ChartQA. I like that the authors investigated different classes of errors and came to some concrete conclusions about why models struggle. It also makes sense to further train models on tasks that might strengthen these areas.

**Summary Of Weaknesses:**

My general sense is that there is not enough in this paper. The paper is short and lacks detail. Since I found the evaluation component most compelling, I wish there was a more in-depth discussion of the findings. The checklist evaluation uses 33 different templates, but the authors only discuss the three broad categories of structural and visual templates, data extraction templates, and numerical QA templates. I wanted to know more about the properties tested by the different templates. I also would have liked to know more about the kinds of errors that models make. Particularly for the numerical tasks, I felt that key evaluation details were lacking. What counts as a failure? If the model rounds differently than the gold answer, is that incorrect? How far are models from the correct answer, typically?

The training section was also lacking key details. Most importantly, it wasn't clear to me whether the models were *pretrained* on the new tasks-- that is, trained from scratch on their original data and these new tasks-- or *finetuned* on the tasks-- that is, the original models were further trained on the new tasks. I know the paper says pretraining, but it is vague about how this was done and doesn't mention reusing the original datasets the models were trained on, which would be necessary in pretraining. The details provided sounded more like finetuning.

I also found the results section in 3.4 very brief. It is puzzling, because the authors take care in selecting the datasets for evaluation, but then provide so little discussion of the actual findings.

---

### Official Review · Reviewer_jqQZ · 2024-09-10

**Overall Assessment:** 4
**Confidence:** 4

**Best Paper:**

1

**Best Paper Justification:**

N/A

**Comments Questions Suggestions And Typos:**

Are remaining DePlot + LLM errors caused mainly by DePlot issues or LLM issues?

**Paper Summary:**

This paper first creates a checklist evaluation suite for ChartQA, visual question answering dataset over chart images. This reveals some weaknesses of models on simple questions, such as looking for presence of certain colors or summing up values. Based on this, the paper then proposes additional continual pretraining tasks to improve the performance of two tested models--MatCha (a fully end-to-end model) and DePlot (a chart-to-table generation model whose output can be fed to an LLM). This yields improvements on the checklist failure cases.

Overall this is a solid paper that blends interesting analysis of error modes with actionable solutions.

**Summary Of Strengths:**

* Identifies error patterns of models on ChartQA using new checklist test set
* Proposes new pre-training tasks to patch these issues

**Summary Of Weaknesses:**

* Does not evaluate larger models, including GPT-4

---

### Decision · Program_Chairs · 2024-09-21

**Decision:**

Accept

**Comment:**

The reviewers found the checklist analysis interested and appreciated that it led to a concrete solution that was implemented. One reviewer commented about the relatively short analysis and expected for more details, including raising many questions that the checklist can answer. The authors would do well to significantly expand their analysis for the final version.